

# Enhancing survivorship and growth of juvenile *Montipora capitata* using the Hawaiian collector urchin *Tripneustes gratilla*

Andrew R. Barrows[1],[*], Joshua R. Hancock[1],[*], David L. Cohen[2], Patrick Gorong[2], Matthew Lewis[2], Sean Louie[2], Lani Musselman[2], Carlo Caruso[1], Spencer Miller[1] and Crawford Drury[1]

[1] Hawaiʻi Institute of Marine Biology, University of Hawaiʻi, Kāneʻohe, HI, United States
[2] Department of Land and Natural Resources, Division of Aquatic Resources, Honolulu, Hawaiʻi, United States
[*] These authors contributed equally to this work.

## ABSTRACT

The biodiversity of coral reef habitats is rapidly declining due to the effects of anthropogenic climate change, prompting the use of active restoration as a mitigation strategy. Sexual propagation can maintain or enhance genetic diversity in restoration of these ecosystems, but these approaches suffer from a range of inefficiencies in rearing and husbandry. Algal overgrowth of juveniles is a major bottleneck in the production of sexually propagated corals that may be alleviated by co-culture with herbivores. We reared juvenile *Montipora capitata* alongside juvenile native Hawaiian collector urchins, *Tripneustes gratilla*, for 15 weeks and documented significant ecological benefits of co-culture. Urchin treatments significantly increased the survivorship of coral aggregates (14%) and individual settlers (24%). We also documented a significant increase in coral growth in the presence of urchins. These results demonstrate the utility of microherbivory in promoting coral growth and survivorship in *ex situ* conditions, providing valuable insight for restoration pipelines of native Hawaiian coral species.

## INTRODUCTION

We acknowledge that the corals, urchins and reefs we study are a part of the ancestral lands of Native Hawaiian people, who have cared for and stewarded these ecosystems for generations.

Global climate change is leading to a decline in coral reef ecosystems (*Hughes et al., 2018*; *Sully et al., 2019*). Rising ocean temperatures are increasing the intensity and frequency of mass coral bleaching events, which are predicted to become a yearly occurrence on most coral reefs by the mid-century (*van Hooidonk et al., 2016*; *Hughes et al., 2017*; *Hughes et al., 2018*; *Sully et al., 2019*). To mitigate global reef decline, managers

Corresponding author
Joshua R. Hancock,
joshhancock92@gmail.com

are increasingly responding through active restoration by directly outplanting sexually and asexually propagated coral stocks onto degraded reefs (*National Academies of Sciences, Engineering, and Medicine et al., 2019*; *Boström-Einarsson et al., 2020*). These initiatives rely on coral aquaculture that produces enough coral material to be ecologically impactful but must be optimized for different species and environments. In Hawaiʻi, where this study was conducted, coral aquaculture and restoration techniques remain fairly new and therefore, understudied despite many threats to the health of Hawaiian reefs (*Forsman et al., 2018*).

Coral restoration initiatives which aim to mitigate the decline of coral reefs have historically relied on asexually derived coral fragments. This "coral gardening approach" requires relatively little maintenance and human effort which makes it a more affordable propagation technique (*Rinkevich, 1995*; *Epstein, Bak & Rinkevich, 2003*). However, this approach requires removing healthy coral which may impact source reefs and results in limited genetic diversity of coral stock (*Rinkevich, 1995*; *Edwards & Clark, 1999*; *Forsman, Rinkevich & Hunter, 2006*; *Lirman et al., 2010*; *Pollock et al., 2017*). Alternatively, sexually derived corals can be reared from gametes providing hundreds of thousands of genotypically diverse corals and offer a more renewable approach to collecting material for restoration (*Raymundo & Maypa, 2004*; *Petersen et al., 2005*; *Guest et al., 2014*; *Barton, Willis & Hutson, 2017*; *Pollock et al., 2017*). Sexually derived corals naturally possess higher levels of genetic diversity, which may dilute genotype-environment mismatches found during outplanting with more genetically depauperate approaches (*van Oppen & Gates, 2006*; *Drury, Manzello & Lirman, 2017*), but see (*Marshall et al., 2010*). This method, however, presents its own logistical and biological challenges. Survivorship bottlenecks are present throughout the rearing process from larval to juvenile life stages in the lab and *in situ* (*i.e.*, benthic competition, predation, bleaching) and take years to reach reproductive maturity (*Wilson & Harrison, 2005*; *Trapon et al., 2013*; *Randall et al., 2020*; *Hancock et al., 2021*; *Rahnke et al., 2022*). Importantly, abiotic factors such as light and flow impact the presence of benthic competitors which affect the survivorship of juvenile corals (*Hancock et al., 2021*).

The success of coral restoration is heavily dependent on abiotic and biotic conditions at an outplant site. Benthic competitors such as crustose coralline algae (CCA), macroalgae, and turf algae may reduce survivorship and growth by shading, smothering, and abrading juvenile corals (*Venera-Ponton et al., 2011*; *Barott & Rohwer, 2012*). High concentrations of turf algae and their associated microbes can also create zones of prolonged hypoxia where respiration outpaces photosynthesis, especially at night (*Silveira et al., 2019*; *Nelson, Wegley Kelly & Haas, 2023*). In combination, these competitive effects may reduce the overall likelihood that any individual coral recruit survives on the reef. Newly settled corals are particularly susceptible to competitive suppression and whole colony mortality. Their small size and limited energy reserves confine their growth, reproduction and defense (*Ritson-Williams et al., 2009*; *Venera-Ponton et al., 2011*; *Tebben et al., 2014*). Coral-algae dynamics are a primary driver of coral reef state and climate disturbances can lead to algal dominance on degraded reefs (*Fung, Seymour & Johnson, 2011*; *Barott & Rohwer, 2012*), although these patterns are region-specific (*Roff & Mumby, 2012*). However, naturally

occurring herbivory can combat these growing effects (*Hughes et al., 2007*). Herbivory reduces algal competition, creating new niche space for newly settled corals and/or providing space for existing corals to grow (*Jompa & McCook, 2002*; *Adam et al., 2015*; *Dang et al., 2020*). Herbivory also plays a crucial role in shaping the structure and composition of reef ecosystems, and understanding these effects on coral can inform the development of *ex situ* rearing methods for restoration of these habitats (*Adam et al., 2011*; *Zarzyczny et al., 2022*).

*Ex situ* aquaculture provides the ability to control abiotic factors (*i.e.*, light and flow) to reduce algal communities while rearing juvenile corals (*Hancock et al., 2021*). Similarly, aquaculture practitioners can leverage biotic interactions (*e.g.*, herbivory) and create space for coral growth (*Toh et al., 2013*). Juvenile herbivores can be particularly effective grazers because of their anatomical compatibility (*e.g.*, small teeth, radula, *etc.*) with juvenile corals. Smaller teeth and grazing appendages likely cause less damage to the corallite structure either incidentally or during grazing at the periphery of the coral. This "microherbivory" is an ecological function which can be performed by many juvenile species such as fish, snails, crabs and urchins. In the Philippines, (*Villanueva, Baria & dela Cruz, 2013*) found that the presence of juvenile herbivorous gastropods, *Trochus niloticus*, increased the survivorship of coral recruits grown in *ex situ* hatcheries. However, *T. niloticus* is cited to only be effective at grazing soft filamentous algae and not at controlling CCA (*Ng et al., 2013*). Similarly, in the Indo-Pacific, (*Craggs et al., 2019*) showed that high densities of *Mespilia globulus* urchins increased the survivorship and growth of *Acropora millepora* by effectively grazing both CCA and turf algae. These studies suggest that coral restoration frameworks can be scaled by leveraging biotic interactions such as herbivory.

To explore the utility of this natural interaction in Hawaiʻi, we employed the native Hawaiian collector urchin, *Tripneustes gratilla*, to investigate the effects of herbivory on juvenile *Montipora capitata* survivorship and growth. We co-reared 180 plugs of newly settled *M. capitata* alongside 135 juvenile *T. gratilla* urchins during the summer of 2021. Our study explores naturally occurring microscale species-species interactions to support coral aquaculture.

## METHODS

### *Montipora capitata* spawning

Juvenile *Montipora capitata* corals were reared following *Hancock et al. (2021)* and *Rahnke et al. (2022)*. Briefly, gametes were collected at Reef 11 (21°26′56″N, 157°47′45″W) in Kāneʻohe Bay, Oʻahu and fertilized on 11 July 2021. Coral gametes were collected under Division of Aquatic Resources (DAR) special activity permit 2022–22 to the Hawaiʻi Institute of Marine Biology. Larvae were allowed to develop for 5 days before settlement on seawater-conditioned (1 week) aragonite plugs. Corals were allowed to settle on 16 July 2021 and allocated into the experiment on 22 July 2021.

## Urchin spawning and rearing

Juvenile collector urchins, *Tripneustes gratilla*, were produced at the State of Hawai'i Division of Aquatic Resources (DAR) Ānuenue Fisheries Research Center (AFRC) Sea Urchin Hatchery Project following best practices outlined in (*Hodin et al., 2019*). Urchins spawned April 5th, 2021 and larvae were reared from gametes collected from wild broodstock in 200-liter cone-bottom fiberglass tanks with filtered, UV treated seawater (FSW) and fed daily with cultured phytoplankton through the larval rearing cycle (22 to 26 days). Competent larvae were transferred to settlement tanks with settlement plates covered by biofilms of benthic diatoms. After 1 week, larvae metamorphosed into juvenile urchins which were placed into grow out tanks after 44 days. Juvenile urchins for this project were harvested from grow-out tanks at AFRC at 87 days old on July 1st, 2021.

## Experimental design

A total of 180 plugs (21 mm diameter) with juvenile coral recruits were randomly assigned to six flow-through 9 L tanks ($n = 30$ plugs per tank, tanks were $30 \times 16 \times 20$ cm with a total of 2,320 cm$^2$ side and bottom area) fed with sand-filtered (#20 silica sand) natural seawater. Each tank was covered with shade cloth and fed with a pump manifold at 120 liters per hour (lph), creating 100% water exchange every ~5 min. This turnover rate created flow within tanks in the absence of an impeller pump, which would have impacted urchins. Light levels were maintained throughout the experiment and averaged 19.6 μmol m$^{-2}$s$^{-1}$ and 16.8 μmol m$^{-2}$s$^{-1}$ in treatment and control tanks, respectively, but there were substantial differences between tanks (range: 9.3–32.0 μmol m$^{-2}$s$^{-1}$). Three tanks were designated as urchin treatments and stocked with juvenile Hawaiian collector urchins, *Tripneustes gratilla*, and three were designated as controls (no urchins). Forty-five urchins, ~1–3 mm in diameter, were added to each treatment tank at the beginning of the experiment, a density slightly higher than *Craggs et al. (2019)*, which used the equivalent of one urchin per plug.

## Image and statistical analysis

Plugs were photographed using a Canon EOS 6D with a macro lens (24–70 mm) to obtain high resolution images of each plug. Photos were taken weekly for the first month, bi-weekly for 6 weeks, then monthly for the final 2 months. Photos were processed in Agisoft Metashape (version 1.8.3) to create a high-resolution orthomosaic image of each rack. Using the 'Draw Polygon' shape tool, we haphazardly selected and traced two aggregates (*i.e.*, multiple coral recruits fused together) and two individual (*i.e.*, discrete) coral juveniles on each plug and calculated area (mm$^2$) of the shape of these corals at initial and final time points using AddTools for Metashape add-on. Orthomosaic imagery of each plug was analyzed at each of 9 timepoints and the number of live individuals and aggregates were counted. For survivorship counts, a coral individual or aggregate was classified as alive if it was pigmented and retained an oral disc visible in the center of the coral. Survivorship was calculated over 105 days and growth was calculated over 91 days.

All statistical analyses were conducted in R 4.2.1 (*R Core Team, 2022*) and data were analyzed separately for individuals and aggregates due to ecological differences between

assemblage type (*Hancock et al., 2021*; *Rahnke et al., 2022*). Aggregated settlers exhibit higher survivorship and lower growth than individuals, likely due to the influence of partial mortality among individual polyps within the aggregation. These effects also influence size and can lead to decreasing size (*i.e.*, negative growth) in early post-settlement stages of this species. We analyzed juvenile survival across the full time-series using the *survival* (*Therneau, 2019*) and *Survminer* (*Kassambara et al., 2017*) packages and compared kaplan-meier curves between treatments. We calculated mean endpoint survivorship per plug and examined the treatment effect using a t-test or wilcoxon rank sum test as appropriate after examining normality assumptions. We calculated percent change in size of individual corals and examined the treatment effect using a wilcoxon rank test. All data and code associated with this project are available at https://doi.org/10.5281/zenodo.8222913.

## RESULTS

Urchins did not injure or kill corals during grazing, but created dramatic differences in CCA and overall algal growth compared to controls (Fig. 1). We tracked survivorship over 105 days among 1,402 aggregate and 1,367 individual corals on 180 total plugs. There was significantly higher survival probability for aggregates and individuals in urchin treatments for the duration of the experiment ($p < 0.001$; Figs. 2A and 2B). Endpoint survivorship was 14.4% higher in aggregates co-cultured with urchins ($p = 0.006$; Fig. 2C) and 24.7% higher in individuals co-cultured with urchins ($p < 0.001$). There was no significant difference in survivorship between aggregates and individuals in the control treatment ($p = 0.790$) or the urchin treatment ($p = 0.257$), although aggregate survivorship was slightly higher than individuals in the control treatment (40% *vs* 36%, respectively).

We calculated growth over 91 days for 230 aggregate and 252 individual juvenile corals (up to $n = 2$ of each type when available, haphazardly selected from each plug) that were alive at the conclusion of the experiment. Individuals and aggregates in control treatments experienced negative growth. The urchin treatment yielded positive growth, which was significantly higher than control growth in both individuals and aggregates ($p < 0.001$; Fig. 3).

## DISCUSSION

Identifying factors that minimize early life history bottlenecks and support juvenile coral success can improve efficiency and reduce limitations of restoration strategies (*Ritson-Williams et al., 2009*; *Craggs et al., 2019*; *Hancock et al., 2021*). Increasing juvenile survivorship remains a critical focus that can provide greater ecological impact and dramatically reduce the cost of creating coral stock for transplanting back onto the reef (*Craggs et al., 2019*). Likewise, tools that increase coral growth should be prioritized, minimizing time to reach project specific goals for size before outplanting more quickly. Here we demonstrate that co-culturing juvenile native Hawaiian collector urchins, *Tripneustes gratilla*, and juvenile *Montipora capitata* effectively increase growth and survivorship of the corals.

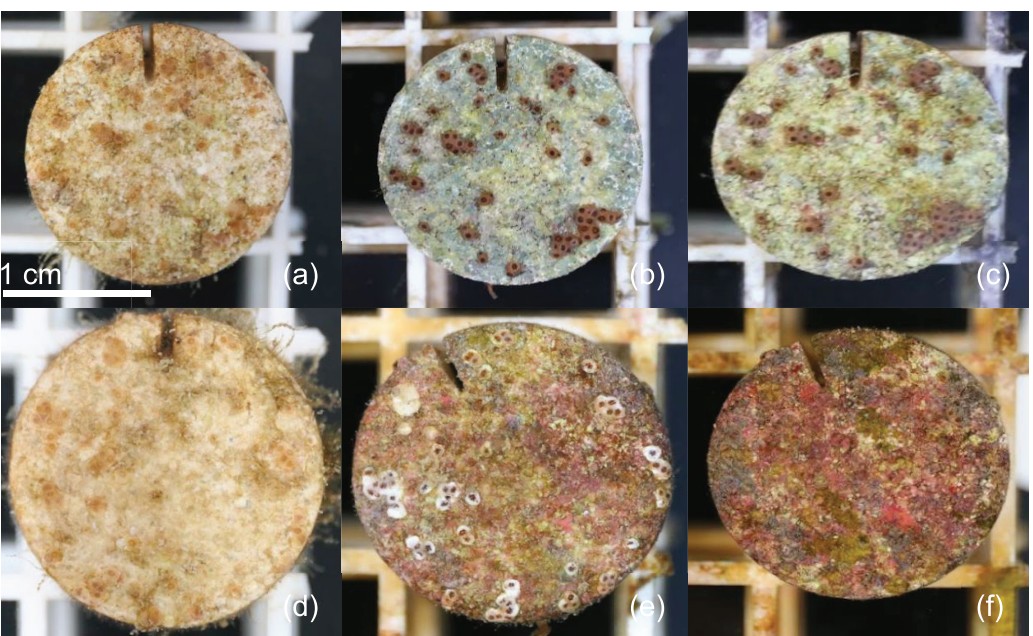

**Figure 1 Example microherbivory patterns.** Timeseries of example plug from urchin treatment at 0 (A), 49 (B) and 105 (C) days post exposure. Timeseries of example plug from control treatment at 0 (D), 49 (E) and 105 (F) days post exposure.

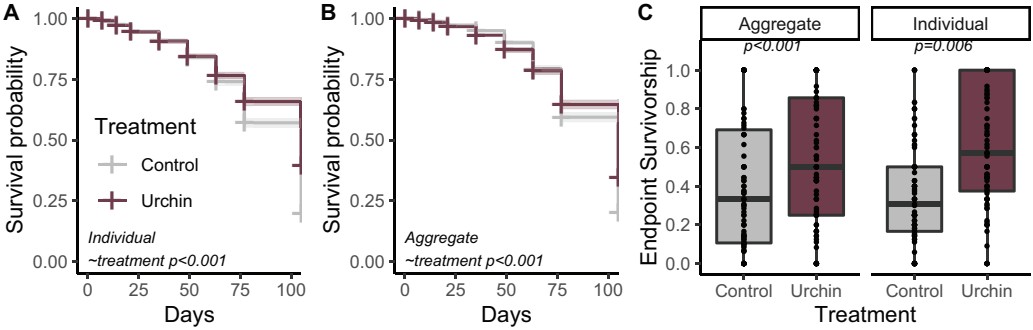

**Figure 2 Juvenile survival patterns.** (A) Survival probability curves through time for individual (B) and aggregate juvenile corals. Colors correspond to treatment. (C) Endpoint actual survivorship for aggregate (left) and individual (right) juvenile corals. Points represent individual plug replicates.

On coral reefs, herbivory plays a crucial role in maintaining ecosystem health and promoting coral abundance (*Littler, Littler & Brooks, 2009*; *Plass-Johnson et al., 2015*). Understanding the effects of herbivory on corals can improve *ex situ* aquaculture methods and restoration outcomes (*Adam et al., 2011*). In our study the presence of juvenile native Hawaiian collector urchins, *Tripneustes gratilla*, resulted in an increase in individual and aggregate survivorship when co-cultured with urchins. This effect is primarily due to the consumption of turf algae by juvenile urchins, similar to (*Villanueva, Baria & dela Cruz, 2013*), who found *T. niloticus* effectively removed turf algae. If left untreated, these turf communities can grow rapidly and smother juvenile corals (*Ritson-Williams et al., 2009*).

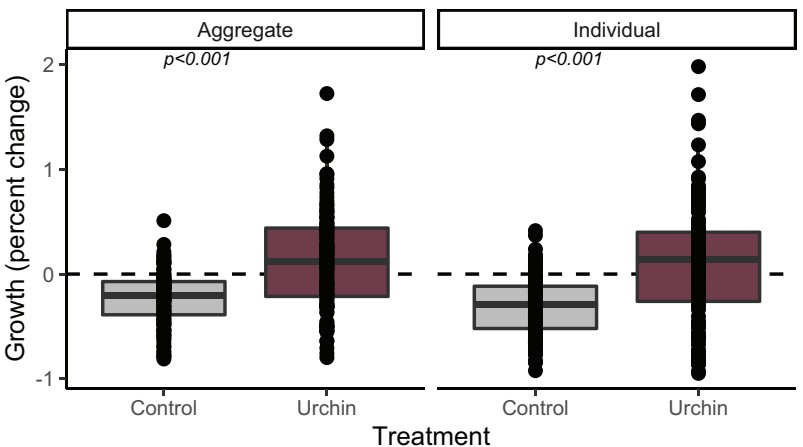

**Figure 3 Juvenile growth patterns.** Relative growth (percent change) for aggregate (left) and individual (right) juvenile corals surviving to the end of the experiment. Colors correspond to treatment.

Algal overgrowth also disrupts heterotrophy and decreases light availability, restricting the photosynthetic efficiency of zooxanthellae, resulting in coral mortality (*Smith et al., 2006*). The light levels in this study were chosen to reduce algal growth but may have also impacted the growth and survivorship of corals around 45 days post fertilization, when juvenile *Montipora capitata* have higher survivorship in higher light regimes (*Hancock et al., 2021*).

Additionally, algal-derived organic matter fosters a high microbial oxygen consumption rate, resulting in enhanced diurnal variability of oxygen concentrations, which ultimately leads to extended periods of nighttime hypoxia that can suffocate corals, promoting a harmful feedback loop. Algae-induced, microbial local hypoxia is one of the main causes of coral mortality in coral-algae interactions (*Nelson, Wegley Kelly & Haas, 2023*).

Early succession of algal communities on coral reefs and in *ex situ* rearing environments is rapid but not immediate. In our study, survivorship in urchin and control treatments was similar for approximately 60 days before the survivorship of the control treatment began to decline more rapidly than the urchin treatment. This suggests a delay in the onset of benthic competition, potentially because plugs used in the experiment were conditioned in seawater for only 1 week prior to settlement. Previous studies have found that this conditioning establishes a beneficial biofilm which attracts settlers, but is not long enough to recruit algal competitors (*Harrington et al., 2004*; *Pollock et al., 2017*; *Hancock et al., 2021*). Thus, juveniles in our experiment did not experience significant competition until algal communities could develop. The consistent grazing of urchins was able to almost entirely prevent the recruitment of algal communities throughout the duration of this experiment, including later successional communities such as CCA. Except for juvenile corals, plug surfaces were stripped bare, resembling the urchin barrens found in temperate kelp forests. We suspect that individuals may receive a greater benefit from microherbivory (24.7% higher survivorship in urchin treatment compared to control) than aggregates (14.4% higher survivorship in urchin treatment compared to controls) as they are smaller

and more easily smothered or abraded by algal interactions compared to larger colonies formed by aggregated settlers. Interestingly, the intermediate disturbance hypothesis, which predicts changes in species richness and abundance within ecosystems based on varying levels of disturbance, may provide additional context for our findings (*Connell, 1978*). We postulate that urchin microherbivory at the density in our experiment represents high disturbance (*i.e.*, physical chewing/scraping) on the benthos. This disturbance was enough to essentially maintain a monoculture, preventing algae while maintaining the coral juvenile.

Benthic competition can have non-lethal consequences for juvenile corals where impaired photosynthesis or heterotrophy and increased energy expenditures affect growth (*Tanner, 1995*; *Potts, 1977*). These impacts were illustrated in our control treatment, where algal overgrowth led to negative growth rates in both individual and aggregate corals. In our experiment, urchin microherbivory resulted in higher coral growth for both individual and aggregate colonies. Our results are consistent with (*Craggs et al., 2019*), who also found that urchin microherbivory benefited *Acropora millepora* growth. Importantly, we did not find evidence that *T. gratilla* damaged or removed coral recruits while grazing, potentially because urchins rely on olfaction and mechanoreceptors in their tube feet for detecting food (*Roberts et al., 2017*), allowing them to distinguish between algae and coral. This specific behavior reduces the prevalence of partial shading and friction/abrasion by turf and crustose coralline algaes, respectively (*Lirman, 2001*; *Jompa & McCook, 2003*). Furthermore, diminished contact with macroalgae may alleviate stress associated with exudates and surface bound compounds, including organic carbon and allelopathic chemicals, which have been shown to alter coral microbial communities and reduce growth (*McCook, Jompa & Diaz-Pulido, 2001*; *Vega Thurber et al., 2012*; *Roach et al., 2020*). Research on, and production of, juvenile corals frequently relies on the manual removal of algae by humans to offset these negative effects, but in our study there was no human intervention, so control treatments experienced notable algal growth and negative growth rates. The lack of human intervention also decreases the cost of sexual production of coral and may eventually help achieve economy of scale for conservation.

Here we show that microherbivory is an effective tool for increasing juvenile coral survivorship and growth, supporting methodologies for programs aiming to increase production of sexually derived coral stock. This work also highlights the importance of collaboration with organizations that specialize in the husbandry of herbivorous species to dramatically improve efficiency of restoration programs. As an integrative approach, these collaborations have the potential to provide financial support for reef restoration, where the cultivation and sale of herbivores used in restoration pipelines create self-sustaining funding models (*Craggs et al., 2019*). Indeed, urchins in particular have been identified as an economically important species and a culturally significant food item in South Korea and the Philippines (*Toha et al., 2017*; *Craggs et al., 2019*). Future research should explore the longer-term impacts of co-culture and how urchin size influences damage to juvenile corals. Importantly, these aquaculture improvements can contribute to increasing the scale of conservation efforts while climate change impacts are addressed at a larger scale.

## ACKNOWLEDGEMENTS

We are grateful to Alyssa Varela, Teagan Roome, Tahirih Perez, Khalil Smith, Kira Hughes and the Coral Resilience Lab for spawning and logistical support for this project. This is HIMB contribution 1939 and SOEST contribution 11723.

### Funding

Joshua Hancock, Andrew Barrows, Spencer Miller, Carlo Caruso and Crawford Drury are funded by the Paul G Allen Family foundation and a grant/cooperative agreement from the National Oceanic and Atmospheric Administration, Project R/SS-34, which is sponsored by the University of Hawai'i Sea Grant College Program, SOEST, under Institutional Grant No. NA22OAR4170108, from NOAA Office of Sea Grant, Department of Commerce. The views expressed herein are those of the author(s) and do not necessarily reflect the views of NOAA or any of its subagencies. UNIHI-SEAGRANT-JC-22-15. David Cohen, Patrick Gorong, Matthew Lewis, Sean Louie, Lani Musselman and the DAR PCSU Sea Urchin Hatchery Project are supported by funding from the National Oceanic and Atmospheric Administration, the United States Fish and Wildlife Service, the State of Hawai'i Department of Land and Natural Resources, and the State of Hawai'i Department of Transportation. The funders had no role in study design, data collection and analysis, decision to publish, or preparation of the manuscript.

### Grant Disclosures

The following grant information was disclosed by the authors:
Paul G. Allen Family Foundation.
University of Hawai'i Sea Grant College Program: NA22OAR4170108.
NOAA Office of Sea Grant.
UNIHI-SEAGRANT-JC-22-15.
National Oceanic and Atmospheric Administration.
United States Fish and Wildlife Service.
State of Hawai'i Department of Land and Natural Resources.
State of Hawai'i Department of Transportation.

### Competing Interests

The authors declare that they have no competing interests.

### Author Contributions

- Andrew R. Barrows conceived and designed the experiments, performed the experiments, analyzed the data, prepared figures and/or tables, authored or reviewed drafts of the article, and approved the final draft.
- Joshua R. Hancock conceived and designed the experiments, performed the experiments, analyzed the data, prepared figures and/or tables, authored or reviewed drafts of the article, and approved the final draft.

- David L. Cohen conceived and designed the experiments, authored or reviewed drafts of the article, and approved the final draft.
- Patrick Gorong performed the experiments, authored or reviewed drafts of the article, and approved the final draft.
- Matthew Lewis performed the experiments, authored or reviewed drafts of the article, and approved the final draft.
- Sean Louie performed the experiments, authored or reviewed drafts of the article, and approved the final draft.
- Lani Musselman performed the experiments, authored or reviewed drafts of the article, and approved the final draft.
- Carlo Caruso conceived and designed the experiments, authored or reviewed drafts of the article, and approved the final draft.
- Spencer Miller analyzed the data, prepared figures and/or tables, authored or reviewed drafts of the article, and approved the final draft.
- Crawford Drury conceived and designed the experiments, analyzed the data, prepared figures and/or tables, authored or reviewed drafts of the article, and approved the final draft.

### Field Study Permissions

The following information was supplied relating to field study approvals (*i.e.*, approving body and any reference numbers):

Coral gametes were collected under Division of Aquatic Resources approved Special Activity Permit 2022–22 granted to the Hawaiʻi Institute of Marine Biology.

### Data Availability

The data and code associated with this project are available at Zenodo: https://doi.org/10.5281/zenodo.8222913.

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
