# Peer review of "Enhancing survivorship and growth of juvenile Montipora capitata using the Hawaiian collector urchin Tripneustes gratilla"

_PeerJ, doi:10.7717/peerj.16113_

## Round 0.1 · original submission · Minor Revisions

I have received evaluations from two reviewers and their insightful comments as well as suggestions for improvement can be seen below. Please ensure that you address all of these comments and suggestions thoroughly in a rebuttal indicating whether or not you agree with each one, what changes have been made and where the change was made in the manuscript.

·

Basic reporting

The paper is professionally written.

The main presentation issue is in Fig 2. The designations in the legend do not match the actual figure panels.

What was the fate of the urchins during the experiment? Was any urchin mortality observed during the experiment and what was the ending urchin size (presumably they grew substantially in 3 mos)?

Experimental design

The experimental design is straightforward and appropriate. However, a few more additional details would be helpful, and possibly, reconsideration of analysis approach. Specifically,
-What are the dimensions of the experimental tanks? The volume is given as 9l, but the scaling of the urchin density would be better related to the grazable surface area (seemingly the area of floor and walls) which requires dimensions. It would also be helpful to articulate how the treatment urchin density (45 per tank) was chosen.
- A bit more details on aggregates vs. individuals would also be beneficial. If I understand correctly, mortality and growth as determined for these two types of entities are fundamentally different things . . . i.e. mortality and growth for an individual represents complete mortality, and growth of a single polyp (either budding or increase/decrease in primary polyp size). For an aggregate, what is reported as ‘growth’ (when negative) likely represents partial mortality; while in order to be scored as ‘mortality’ it requires death of multiple polyps. I realize that Aggregates and Individuals are analyzed completely separately (which is appropriate given the distinctness noted above). Given this, it is sort of interesting that the endpoint survivorship was quite similar, but I think some mention of partial mortality is warranted.

Validity of the findings

Results of the study are consistent with prior work on herbivore co-culture.
Relative to point above about individuals vs. aggregates: Ln 216-219 requires additional consideration/explanation. I am not clear the derivation of the stated 6% improvement in survivorship . . . it seems to me that the expected improved survivorship of aggregates (over individuals) is a bit tautological (i.e. more polyps have to die in order to be scored as ‘mortality’). It states ‘this effect was not significant’, but I didn’t think any statistical comparisons were conducted between aggregates and individuals . .

Additional comments

Ln 89-90: more explanation of this assertion is needed (what is meant by anatomical compatibility with juve corals’?)
Ln 92: The cited study did not occur in the Caribbean (rather Philippines)
Ln 182: Why is ‘cost’ in quotes? This does not seem like a colloquial expression (as I generally take quote marks to indiciate)
Ln 183: Is the ‘transplantable size’ a known and established quantity?
Ln 247-257: The authors’ idea that partnership with commercial (I presume this means food-producing) urchin culture could leverage costs of restoration is an interesting one. Ln 255-257 implies that the present study featured such leverage (i.e., reduced cost/labor based on collaboration with aquaculture specialists). Is the urchin culture facility sourced in the present study a food-producing, commercial facility? It sounds like a research/restoration oriented facility
In the reviewing .pdf, the ref list appears twice

·

Basic reporting

no comment

Experimental design

no comment

Validity of the findings

no comment

Additional comments

Review of Barrow et al Peerj #86642 Enhancing survivorship and growth of juvenile Montipora capitata using the Hawaiian collector urchin Tripneustes gratilla


The manuscript by Barron et al. titled “Enhancing survivorship and growth of juvenile Montipora capitata using the Hawaiian collector urchin Tripneustes gratilla” reports the successful co-culture of two Hawaiian reef species, the coral Montipora capitata and sea urchin Tripneustes gratilla. Following ex situ spawning, and initial grow out of T.gratilla in April 2021 and the in situ collection of M.capitata gametes during wild spawning in July 2021, the authors co rear these two species over a 105 day period. Survival and growth of the coral were shown to be enhanced as a result of co culturing compared to corals raised in the absence of grazers (control).

The manuscript has been well written, with a clear, well-structured introduction that covers the rationale behind the experiment, along with appropriate literature references within the context of the study. An excellent discussion, well contextualized within the literature. In addition, the study builds on previous co culturing experiments by evaluating the role of coral survival and growth, as a result of spat growing either individually or within an aggregated multi genotype entity.

I would like to see more detail is certain parts of the manuscript to enable other practitioners to utilise this approach within their own work and so have provided detail of these recommendations below.
The figures, tables and RAW data included are all relevant to the manuscript and the experimental design and statistical models use are appropriate.

Suggested improvements are below:
Line 115 - 124 – Considerably more detail about the approach used to spawn and rear the Tripneustes is needed. The authors provide two references (Hancock et al 2021 and Rahnke et al 2022) that each have clear methodology to follow for the coral rearing. If a paper exists that covers the urchin rearing in similar detail please reference this. If not then expand, to provide information of broodstock conditioning diet, environmental parameters that the broodstock are maintain, method of spawning and fertilisation, great detail of the larval rearing vessel, the species of phytoplankton used to rear the echinopluteus, phytoplankton feeding densities and larval settlement tanks etc. This whole paper highlights the benefits of this approach to increase the survival, growth and therefore potential scale at which corals can be reared. Therefore for other practitioners being able to utilise this and achieve similar results they will need to able to replicate the coral and urchin rearing approaches.

Line 129 – I assume this is a flow through system utilising natural seawater (NSW) for exchange rather than a closed recirculating ex situ system. If so, please clarify. Information such as the level of micron filtration the incoming NSW would be useful to understand, as this will influence the amount of naturally occurring feed availability that will influence coral heterotrophy.

Line 131 – Were light levels maintained at this level throughout the 105 day experiment? It seems an appropriate level for the first month while the spat are acquiring symbionts but rather low as they become established over the following months.

Line 135 – can you provide the urchin density and what was the rational behind
adopting this density? Ogden et al (Ogeden, N. Ogden, J. C., & Abbott, I. A. Distribution
abundance and food of sea urchins on a leeward Hawaiian reef. Bull. Mar. Sci. 45, 539–549
(1989)), showed a peak densities of adults of 73.6 m2 in Hawaii which was formed the rational behind the high density treatment in the Craggs et al 2019 paper.

Line 165 & 166 – It maybe the way I’m reading the figure but the end point % survival in Fig 2 A&B don’t seem to cross reference to C (with A&B having lower rates compared to C. Also see comment later on Figure 2 legend.

Line 171 – Can you clarify here. If you started with 1402 aggregate and 1367 individual corals (Line 163) but have 230 aggregate and 252 individual juvenile corals at 91 days this would provide a survival rate of 16.41% for aggregate and 18.44% for individual no? This seems to contradict the results discussed in line 165 – 170.

Line 241- 244 – I can see what is trying to be said here but perhaps expand on this to highlight how labor reduction as a result co culturing leads to a decrease in grown out cost. It is through utilising there nature-based solutions that economy of scales can be achieved.
Line 263 – 264. This line needs finishing xx

Line 288 – 291 Figure 2 legend needs rewriting as it doesn’t represent the figure. There’s no reference to fig C and the left and right description for A & B are the wrong way round.

Overall, I think this is a nice manuscript that adds to the literature about the importance of a multi trophic aquaculture approach to enhance coral production for restoration. Having visited HIMB and AFRC a while back there is an excellent opportunity to continue to develop this collaborative work and exploring if the benefits of this approach that are seen ex situ, continue to apply if juveniles corals and urchins are out-planted together in sea cages for example.

---

## Round 0.2 · accepted · Accept

I am satisfied with the changes that have been made to the manuscript and in my opinion, the manuscript is acceptable for publication in PeerJ. Thank you for your contribution.